# Fabrication of a Reflective Optical Imaging Device for Early Detection of Breast Cancer

**DOI:** 10.3390/bioengineering10111272

**Published:** 2023-11-01

**Authors:** Huu Thuan Mai, Duc Quan Ngo, Hong Phuong Thi Nguyen, Duong Duc La

**Affiliations:** 1School of Engineering Physics, Hanoi University of Science and Technology, No. 1 Dai Co Viet, Hanoi 100000, Vietnam; quan.ngoduc@hust.edu.vn; 2School of Chemical Engineering, Hanoi University of Science and Technology, No. 1 Dai Co Viet, Hanoi 100000, Vietnam; phuong.nguyenthihong@hust.edu.vn; 3Institute of Chemistry and Materials, 17 Hoang Sam, Hoang Quoc Viet, Cau Giay, Hanoi 100000, Vietnam

**Keywords:** BKA-06, blood vessels detection device, red LED lights, medicine

## Abstract

This work presented the design and fabrication of a blood vessel and breast tumor detection device (BKA-06) based on optical energy spectroscopy. The BKA-06 device uses red-to-near-infrared light-emitting diodes that allow physicians or physicians to visualize blood vessels and surface structures such as breast tumors with the naked eye. The device consists of a built-in current control circuit to have the appropriate brightness (maximum illuminance of 98,592 lux) for the examination of superficial tumors deep under the skin, with a scan time of 3–5 min. The device BKA-06 can facilely observe each layer of blood vessels at the depth of the skin. For breast tumors, the location, size, and invasive areas around the tumor can also be visualized with the naked eye using the BKA-06 sensor. The results show that the BKA-06 sensor can provide clear breast tumor and vascular images, with a penetration of up to 15 cm in the skin and tissue layers of the breast. The breast tumor scanning tests with the BKA-06 sensor gave patients quick results and compared them through cell biopsy and MRI, respectively. The device has the advantages of being simple and easy to use, providing potential practical applications in the medical field and reducing costs for patients when taking MRI or CT scans. Therefore, the BKA-06 device is expected to help doctors and medical staff overcome difficulties in infusion, as well as identify breast tumors to support early breast cancer diagnosis and treatment.

## 1. Introduction

In medicine, the skill of identifying veins in infusions and taking blood is a skill that not all doctors can perform well. Finding a vein is difficult (especially for children, women, and obesity) [1]. It is not uncommon for cases of vein deviation or the rupture of vessels when injected or drawn blood that has adversely affected the patient. The vein retriever process requires the precise operation of people with experience that are not allowed to make mistakes because, if performed many times, will cause pain and discomfort to the patient [2].

On the other hand, the problem of breast cancer shows that the incidence of breast cancer is increasing in many parts of the world, especially in transition countries [3]. Sadly, an estimated 685,000 women lost their lives to breast cancer in 2020, making up 16% of all cancer deaths in women or 1 in every 6 deaths. To address this concerning trend, the World Health Organization (WHO) recently initiated the Global Breast Cancer Initiative [4]. In 2018, Vietnam ranked first in the world in terms of breast cancer incidence, with 15,229 new cases and 6103 deaths. In 2020, there were over 21,555 new cases, including more than 9345 deaths because of breast cancer [5]. Currently, Vietnam annually records about 183,000 new cases of cancer, of which breast cancer accounts for 25.8% of cancers in women, with nearly 22,000 new cases and more than 9000 deaths [6]. Notably, often, the younger breast cancer patients are, the worse the prognosis and the lower the cure rate compared to the elderly. Epidemiological studies indicated that the most effective method to increase the cure rate of breast cancer, especially in young people, is screening examination to detect the disease early. While the over 5-year survival rate for the very early stage reaches 98%, for the late stage, this rate is only about 10%. This collaborative effort aims to reduce breast cancer mortality by promoting prompt diagnosis, effective treatment, and comprehensive patient management. Currently, there are many advanced diagnostic techniques, such as the BRCA1/2 gene test, breast ultrasound elastography, 3D mammogram, magnetic resonance imaging (MRI), breast tumor biopsy, ultrasound, computed tomography (CT), etc.

Past studies have indicated that the timely detection and appropriate treatment of breast cancer could significantly decrease mortality rates in the long run [7]. At present, there is a wide array of clinical approaches employed for the screening and diagnosis of breast cancer. Although mammography is currently the standard screening method for breast cancer, it has limitations. It is less effective for individuals under 40 years old and with dense breasts, and it is not sensitive enough to detect small tumors (less than 1 mm, approximately 100,000 cells). Furthermore, it does not provide any information on the potential outcome of the disease [8,9]. Contrast-enhanced (CE) digital mammography, on the other hand, offers a more precise diagnosis than mammography and ultrasound for individuals with dense breasts. However, this technique is not widely accessible due to its cost and the high levels of radiation involved [10]. Ultrasound has been utilized as an additional tool for screening. Magnetic resonance imaging (MRI) has the capability to identify small lesions that may not be detectable through mammography, but it is also costly and has low specificity, potentially leading to overdiagnosis [9,10]. Positron emission tomography (PET) is recognized as the most accurate method for visualizing tumor spread or monitoring their response to treatment [11].

However, the aforementioned methods face challenges when it comes to widespread implementation in early-stage cancer screening due to their limited availability in local medical facilities, high costs, lengthy result turnaround times, and complex processes [12]. On the other hand, thermal imaging has emerged as an effective method for the early detection and diagnosis of breast cancer. Reflectance optical imaging operates by quantifying both transmitted and reflected photons and utilizing this data alongside tomography reconstruction algorithms to create images of spatial absorption and/or scattering patterns [13,14]. These optical imaging systems have the capacity to generate three-dimensional (3D) images [15]. In recent decades, efforts have been made to advance these techniques for clinical research [16]. However, the complexity and high expenses associated with time domain (TD) systems have impeded their widespread adoption [17]. On the other hand, frequency domain (FD) systems function by modulating the light source’s amplitude at a high frequency (>50 MHz), with detectors measuring the reduction in amplitude and phase shift of the transmitted signal [18]. Conversely, continuous wave (CW) systems emit near-infrared (NIR) light at a consistent intensity or with low-frequency modulation (a few kHz) to enhance the signal-to-noise ratio. In CW systems, detectors gauge the reduction in amplitude of the transmitted signals [19]. CW systems have been enhanced with multiwavelength laser diodes and dense arrays of sources and detectors, making them valuable in clinical research [20,21]. CW systems only require photo detectors with a slow response rate and low-frequency circuits. Consequently, CW systems are significantly more cost-effective and less complex than FD systems, allowing for the integration of more sources and detectors [22]. Nevertheless, relying solely on amplitude data from CW systems presents limitations in quantitative optical reconstruction, and clinical applications have predominantly focused on monitoring dynamic changes in breast abnormalities [23]. To address this limitation and strike a balance between cost and complexity, FD and CW systems have been combined [24].

In this study, we have focused on the development and testing of the CW light-based reflectance device (named BKA-06 device), which simulates the energy absorption image of cancer (particularly breast cancer and superficial cancers). BK06 can provide valuable contribution to the field by addressing the limitations of the optical reflectance imaging method, such as enhanced accuracy, providing real-time results, cost-efficient, and less harmful to tissues. Through testing on both volunteers and patients, this device has demonstrated outstanding advantages compared to similar devices, such as (1) cheap price (250–300 USD/device); (2) to give fast results (in 3–5 min); (3) no side effects; (4) to produce a clear and accurate image and location of the tumor within a range of less than 15 cm (compared with MRI). Additionally, the device also aids in identifying blood vessels during patient treatment, which is crucial for procedures such as blood collection and injection/infusion. It is worth noting that not all medical professionals possess the expertise to accurately identify blood vessels, and misidentification can have detrimental effects on patients, especially children.

## 2. Materials and Methods

### 2.1. Research Methods

BKA-06 is based on the biomedical physics principles. Analyze and select electronic elements and suitable materials for fabrication equipment.

### 2.2. Physical Basis and Operating Principle of BKA-06

In the composition of red blood cells, there is a heme pigment, the porphyrin type that may combine with metal atoms. Heme in humans is protoporphyrin IX in combination with Fe. Heme has 4 pyrol nuclei linked together by the menten bridge (-CH=). Porphyrin rings are attached to the groups as metyl (-CH_3_) at the sites 1, 3, 5, and 8; vinyl (-CH=CH_2_) at the sites 2 and 4; and propionyl (-CH_2_-CH_2_-C00H) at the sites 6 and 7.

Within heme, the iron atom is attached to the pyrrole nucleus by two symmetrical and two coordinated links, as well as to the protein globin through a histidine residue at the base. These bonds are crucial for the functioning of hemoglobin (Hb) (Figure 1), as they help to stabilize the structure and allow for the reversible binding of oxygen.

Each Hb molecule consists of four hemes, with one heme containing 1 Fe^2+^ Spectrometry analysis of blood with the maximum absorption spectrum within the 540–700 nm region, with the maximum absorbance of the SHb component at 622.8 nm [11,25]. Human blood with a hematocrit of 10% and oxygen saturation of 98% displays optical properties of 0.210 ± 0.002 mm^−1^ for μa, 77.3 ± 0.5 mm^−1^ for μs, and 0.994 ± 0.001 for the g factor [26]. As hematocrit increases up to 50%, there is a linear increase in absorption and a decrease in scattering. Variations in osmosis and the wall cutting speed affect all three parameters, while oxygen saturation has a significant impact solely on the absorption coefficient. Measurements of oxidized and deoxidized blood absorption spectra in the 400–2500 nm wavelength range demonstrate that blood absorption follows that of hemoglobin and water. The scattering factor for λ = 500 nm reduces to approximately 1.7, and the g coefficient is 0.9 higher for the entire wavelength range [17].

### 2.3. Techniques

Analysis of the circuit components and electronic devices.

Determine the physical parameters of the device by modern measuring devices and reliable devices such as: Kyoritsu 1052 (Kyoritsu, Tokyo, Japan), Lux Meter Testo 0500 (Testo, PA, USA), and Testo 608-H2-US electronic thermometer (Testo, PA, USA).

### 2.4. Fabrication Design

Component selection: RED LIGHT LED Chip, IC Stable Pressure, and Line Stabilization.

Line stabilization circuit design for LED as shown in Figure 2.

The equipment shell of the device is exhibited in Figure 3.

### 2.5. Measurement Process

Experiment 1: After designing and testing the elements of the device, the equipment is assembled according to the design scheme, and then, the basic electrical parameters of the BKA-06 (voltage and current intensity when operating) are measured using a universal meter (Kyoritsu 1052—Japan) and some other indicators.

Experiment 2: Survey the parameters of BKA-06: measure the brightness intensity using the three modes of the device (start, shallow test, and deep test) and check the heat increase of the device in the screening area for a period of 1 to 25 min.

Experiment 3: Screening of BKA-06 on the surface of the hands of adults and children.

Experiment 4: Use BKA-06 to identify the position on volunteers and breast cancer patients and then test with MRI.

The results obtained from both patients and volunteers have been officially certified with ethics certificate number V11.CN6.3000.23 by the Vietnam Metrology Institute.

## 3. Results

### 3.1. Studies of Basic Parameters of BKA-06

#### 3.1.1. Device Specifications

Figure 4 shows the printed circuit scheme, BKA-06 device, and schematic layout of the device with following specifications:

Changing power source: 220 V~50 Hz, power supply: lithium–ion battery of 9 V

Power Consumption: 12.19 W

Dimensions: 210.08 × 41.22 × 50.08 mm; Φ 32.92 ± 0.02 mm (use caliper, micrometer)

Weight: 892 g

The ability to penetrate the layers of tissue in the breast is 15 cm.

Device BKA-06 (early vein detection and breast cancer detection device by the energy spectrum) has been licensed by the Metrology Institute—Vietnam Academy (attached to certificate no.: V11.CN6.300.23).

#### 3.1.2. Measure Brightness

Using the lux meter (Testo 0500, Testo, PA, USA) and multimeter (Kyoritsu 1062, Kyoritsu, Tokyo, Japan), place the sensor head of the lux meter close to the header of the BKA-06 equipment and measure the brightness intensity in different changing modes. The result is shown in Table 1 and Figure 5.

The survey results show that the amperage reaches 0.75 to 2.80 A, the maximum power consumption is 12.19 W, and the illuminance increases from 5.03 to 143.93 W/m^2^. With an illuminance from 5.03 to 21.73 W/m^2^, which is bright enough to examine capillaries close to the skin, this mode is suitable for superficial tumor surfacing or vascular examination with the BKA-06 device (Figure 6b,c). With an illuminance from 28.08 to 91.31 W/m^2^, suitable for the vascular examination of tumors and superficial subcutaneous lymph nodes (neck, nasopharynx, and breast), this mode has a penetration depth of about 3 ÷ 7 cm (Figure 6a). The illuminance from 91.31 to 143.93 W/m^2^ is suitable for deep lymph nodes under the skin of the breast 7 ÷ 15 cm for breast cancer diagnosis (using the Sony Alpha ILCE-6400L/A6400 Kit and 16–50 mm F3.5–5.6 OSS Camera).

### 3.2. Vascular Results in Adults and Children

An angiogram of the blood vessels in the arm and hands (6 adults) is shown in Figure 6b,c.

The obtained results show that the images of the blood vessels under the skin are very clear, and their positions are easily recognized with the naked eye.

The results obtained show that the images of blood vessels under the skin are very clear and easily identifiable with the naked eye when using the BKA-06 device. Because the BKA-06 device heats the area for imaging, the thermal camera also provides clear images of the blood vessel locations. Additionally, for blood vessels located deeper at 2 cm to 3 cm, heating with the BKA-06 also yields images with clearer contrast of the blood vessels on the thermal camera. The maximum temperature increase of the device during scanning is ΔT °C < 1 °C; this temperature increase is very small compared to the change in body temperature according to environmental temperature. The comparison of the images clearly indicates that the contrast of blood vessel images by the BKA-06 is more distinct.

### 3.3. Breast Test Results on Volunteers and Patients

#### 3.3.1. Volunteers

Volunteers were randomly selected between the ages of 18 and 60 (never had breast cancer tested with any medical devices, no suspicious or conclusive signs of breast cancer). The initial scan results are shown in Figure 7 and Figure 8 and Table 2.

Figure 9 clearly shows the non-screening area and the scanning area. At the illuminated area, we observe with the eye the mammary gland blood vessels with a dark red color, and with the appropriate brightness intensity, we can observe the layers of breast tissue. Therefore, there is no need for an infrared camera auxiliary device to connect a computer to simulate the image, as we can also see the inside image of the breast when using the equipment.

The results of breast cancer screening with the BKA-06 device on 15 volunteers showed quality images, observed with the naked eye; the mammary glands and blood vessels were very clear, with no abnormal organization and no lymphoma.

#### 3.3.2. Patient Participation

Twelve (12) patients who took part in the scan had abnormal breast abnormalities. Two MRI patients were found to have tumors, then had breast cancer scans using BKA-06 devices; the rest were clinically examined and suspected of a tumor from a MRI scan, so conducted a BKA-06 scan, MRI scan, and cell biopsies. 

Results:

+ Patient profile: Patient No. 6, age of 58. Profile number: 7673/20. Diagnose: Mammary tumor (T), size 60 × 63 mm, middle tumor in the breast (T), non-mobile soft tumor (Δ(t): left mammary K (T) T_4_N_0_M_0_. GTB UT GDII). The MRI image shows the tumor, location, and size of 60 × 63 mm of these images taken after the photoreceptor injection and through a computer simulation system. Observing two perpendicular images from the BKA-06 device, we also see that the tumor area is darker than the area around the tumor; the border area is especially very clear, so it is easy to recognize the shape of the tumor’s position and size. Because the device has not been integrated with a computer, the tumor size on the image of BKA-06 is not quantized, but through the naked eye, we also estimate the size of the tumor. The results are shown in Figure 9.

+ Patient profile: Patient No. 7. Profile number: 7692/20. Diagnose: Mammary tumor (T), size 16 × 29 mm, inner protrusive bottom (T), I am probably not mobile (Δ(t): left mammary gland (T) T4N0M0. GTB UT GDII). Results of MRI scans, BKA-06, and cell biopsies are shown in Figure 10.

Similar to patient 6, the images taken of the disease give clearer and darker images (it may be because one patient has a different type with solid lymph nodes, according to the results of the clinical examination) so we can observe the depth of the tumor; the area around the tumor (yellow vinculum) develops many capillary organizations, so it is darker than the middle area tumor center (blue vinculum in Figure 11). The cell biopsy results show melanoma. The results on the MRI show the location and size of the tumor (16 × 29 mm).

The results of the BKA-06, MRI, and cell biopsies are summarized in Table 3. Observations from the images taken with BKA-06 shows that the location and size of the tumor is very clear and with a shallow depth or same cell density.

The results of the MRI scans, BKA-06, and cell biopsies are exhibited in Figure 12. Observations from the images taken using the BKA-06 show that the location and size of the tumor are very clear, no matter the depth or density of the cells and the internal environment in and outside the typical tumors in patients UHV1, UHV7, UHV8, and UHV9. On the other hand, the upper breast MRI images in Figure 12 from the patients also show the location and size of the tumor with the same results as BKA-06. The advantage is that the penetrating image compared to the black and white image on the MRI is recommended to observe the tumor boundary area.

## 4. Discussion

The BKA-06 device obtains 2D images of the screening area on the body. Visually observing the images with the naked eye, we can see some of the structures inside the skin that are layer-by-layer vascular systems in depth [27,28,29]. For breast tumors, it is also close to the location, size, and invasive areas around the tumor.

The device has the advantages of a simple structure, easy to use, and clear images that can be observed with the naked eye. The area of light emitted in the red area of the spectrum is not harmful to the eyes or skin, because infrared lights are used in the treatment of skin diseases in the cosmetic industry and other areas in health.

The initial results of the successful manufacture of BKA-06 and some potential results can be upgraded to create new essential products for the medical industry, improving the quality of infusions and blood draws.

BKA-06 is specialized for a blood angiogram of shallow tumors. There are outstanding advantages when it comes to breast scans compared to CT scan or MRI methods. Without photovoltaic injections, the 3 ÷ 5 min scan time (CT scan, MRI 25 ÷ 30 min) costs an estimated 7 ÷ 9 million USD, so if the cost of a test is negligible compared to the MRI (2.3 ÷ 2.8 million USD/1 degree), the patient can take a selfie and easily see their results, making it easy to use and with a low cost about 250–300 USD/1 BKA-06 device.

BKA-06 has not been able to capture internal organs, deeper locations, and bone organizations that can be taken by MRI or CT scans, and deep positions on breasts larger than 15 cm cannot be observed.

The theory successfully helped design BKA-06 and initially achieved some good results that can be upgraded to create new essential products for the medical industry and improve diagnostic methods for medical facilities on the domestic and international levels.

The imaging method based on the principle of energy absorption of the mass is a new method in diagnostic imaging, both in the world and in Vietnam, so this study could help to access and update research and applications in other industries. The BKA-06 device is also expected to have superior applications with other diagnostic methods. The BKA-06 device is also expected to have applications that are superior to reality.

The BKA-06 has many advantages, such as technological advancements: The device is capable of providing high-resolution, real-time images of breast tissue, allowing for better visualization of potential abnormalities; non-invasiveness: Unlike traditional mammography, it does not involve ionizing radiation, making it a safer option for regular breast cancer screening; cost-effectiveness: Reflectance optical imaging has the potential to be a cost-effective breast cancer detection method, especially in comparison to more expensive imaging modalities like MRI; and patient comfort: Patients may find reflectance optical imaging more comfortable and less intimidating than traditional mammography, which could encourage more regular screenings.

However, for clinical application, it might also pose several challenges, such as clinical validation (large-scale clinical trials are needed to establish its accuracy and reliability), limited depth penetration (this can affect its sensitivity in detecting tumors located deeper within the breast), operator training (effective use of these devices requires specialized training for healthcare providers), false positives and negatives, integration into the clinical workflow, and reimbursement and regulations (regulatory approval and reimbursement policies need to align with the adoption of this technology. It may take time to navigate these aspects.). Addressing the challenges related to clinical validation, operator training, and integration into clinical practice will be critical for its successful implementation as a routine screening tool for breast cancer. Ongoing research and collaboration between healthcare providers, researchers, and regulatory agencies are essential to overcome these challenges.

## 5. Conclusions

In short, we successfully designed and fabricated equipment (BKA-06) with basic specifications: power consumption of 12.19 W, size of 210.08 × 41.22 × 50.08 mm, Φ 32.92 ± 0.02 mm, and weight of 892 g. The device is capable of penetrating the layers of tissue on the breast up to 15 cm. BKA-06 does not burn during screening, emits red light (the main wavelength of 633 nm is healthy radiation and does not cause side effects), there are three screening modes, the maximum brightness intensity is 98,238 lux, inspection time is 3 ÷ 5 min, and the image is visible to the naked eye. We have reviewed the safety specifications when using the device in various modes (starting, shallow test, and deep test mode) during venipuncture, as well as breast scan examinations. We have successfully tested the device on volunteers and patients, with very good initial results. The XCS observations identified blood vessels, tumors in the breast, and surrounding organizations very clearly to determine the location, size of the tumor, and the area around the tumor directly without surgery or other diagnostic methods. A parallel examination using MRI and cell biopsies were performed, with reliable results. In future work, we will continue the testing of patients on a larger scale to evaluate the reliability of the device, as well as optimizing the processing factors of the device, to obtain reliable results for breast cancer detection.

## Figures and Tables

**Figure 1 bioengineering-10-01272-f001:**
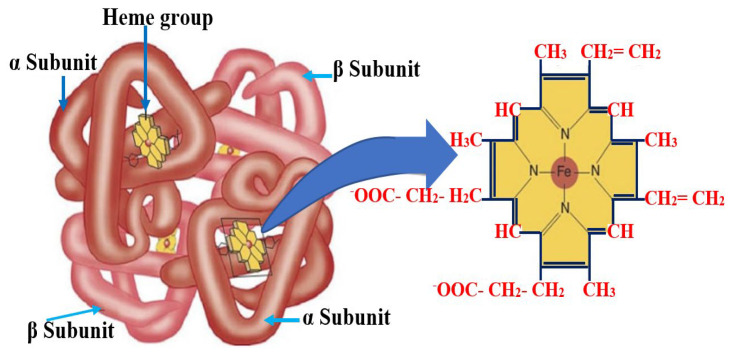
Chemical formular and structure of isoleucine and hemoglobin—the nucleus of red blood cells.

**Figure 2 bioengineering-10-01272-f002:**
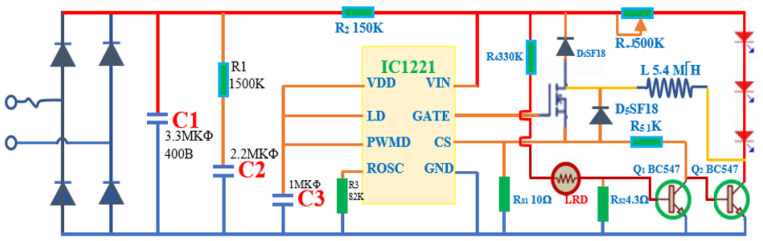
Current stabilizer circuit for LED and sensor (the red line is the positive wire, the blue line is the ground wire, the orange line is the neutral wire; D1 to D4 are the 1N4001 diodes, C1 to C3 are the capacitors; R refers to the resistor; Q refers to the transistors; L refers to the coils; D5 to D7 are the red light-emitting diodes).

**Figure 3 bioengineering-10-01272-f003:**
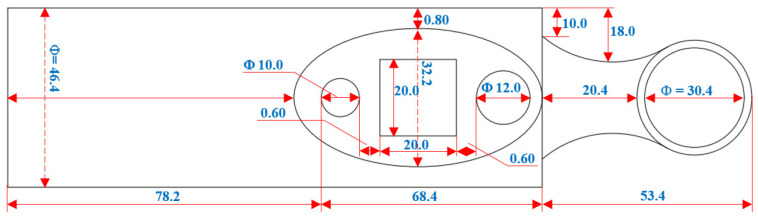
Device shell scheme. Device case: aluminum. Projector material: hard plastic. Projector diameter: Φ = 32.90 mm. Projector length: 34.50 mm. Device size: 200.00 × 38.20 × 50.06 mm.

**Figure 4 bioengineering-10-01272-f004:**
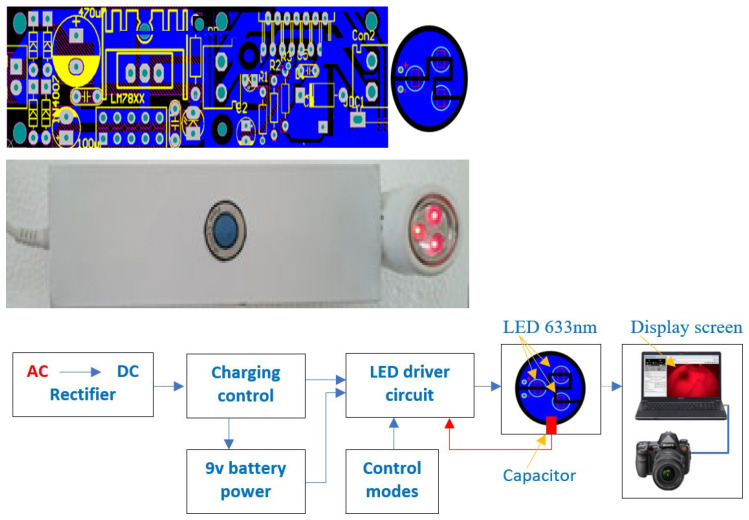
Printed circuit scheme, BKA-06 device, and schematic layout of the device.

**Figure 5 bioengineering-10-01272-f005:**
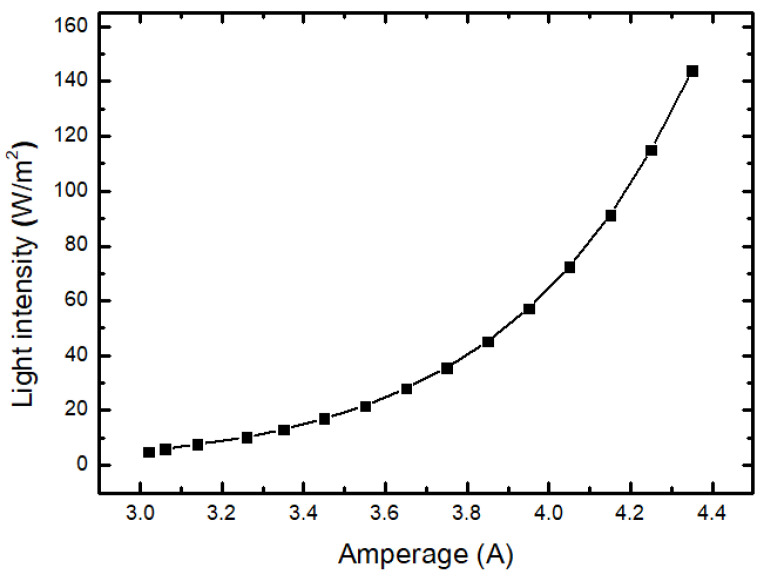
Relationship between amperage and illuminance.

**Figure 6 bioengineering-10-01272-f006:**
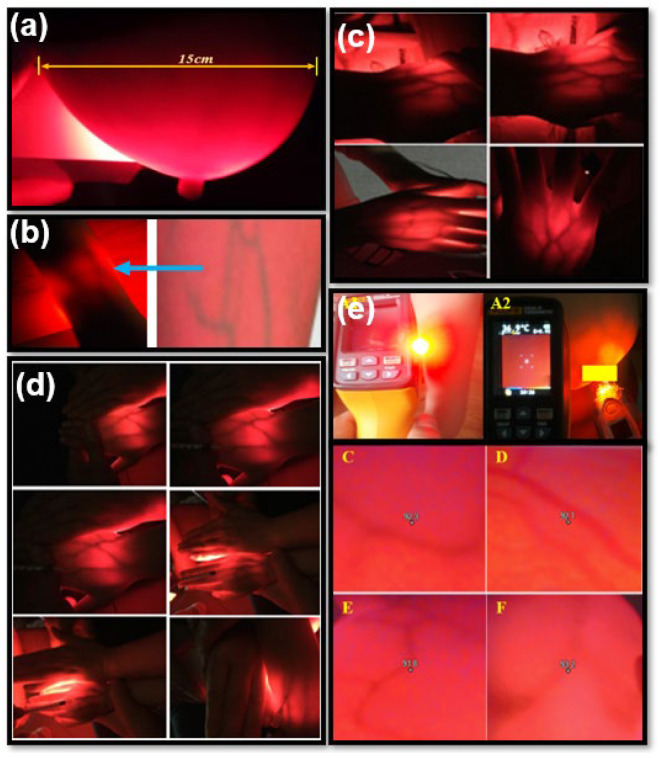
(**a**) BKA-06 equipment image that scans through layers of adult breast tissue. (**b**) Vascular screening on the surface of an adult’s arm using BKA-06 equipment. (**c**) Vascular screening on an adult’s hand using BKA-06 equipment. (**d**) Vascular screening on a child’s hands using BKA-06 equipment. (**e**) The thermal images of blood vessels illuminated by BKA-06 on (A1, C, and D) a child’s forearm and (A2, E, and F) adult’s breast.

**Figure 7 bioengineering-10-01272-f007:**
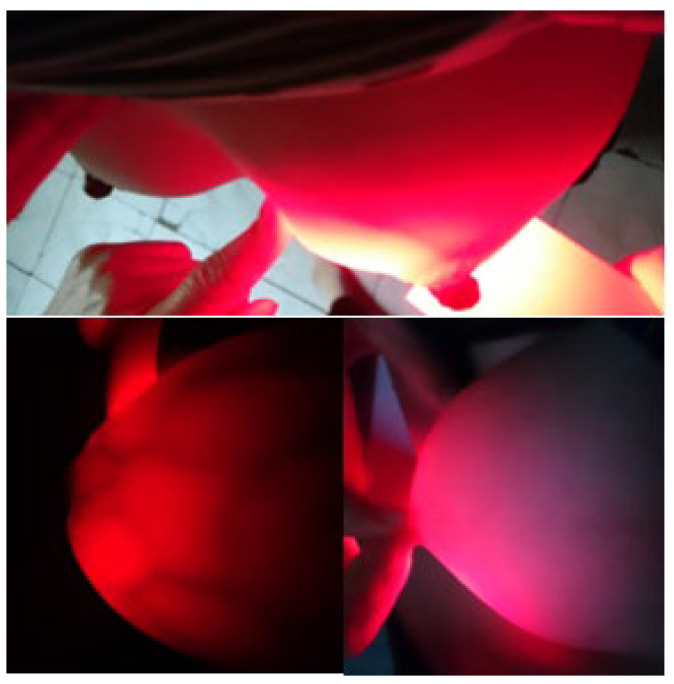
Breast test photos of volunteer no. 9.

**Figure 8 bioengineering-10-01272-f008:**
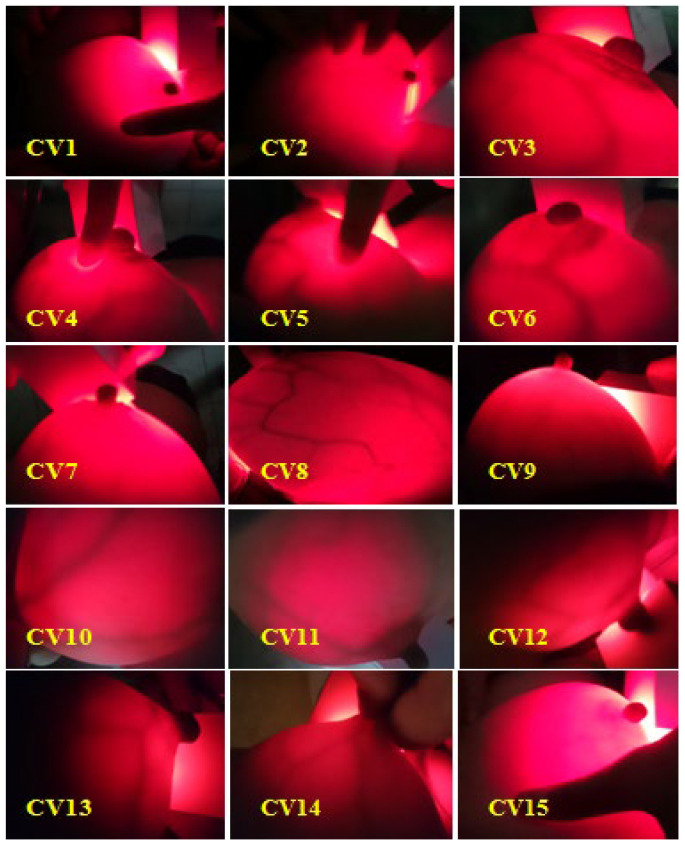
Summary of breast cancer test results using the BKA-06 device.

**Figure 9 bioengineering-10-01272-f009:**
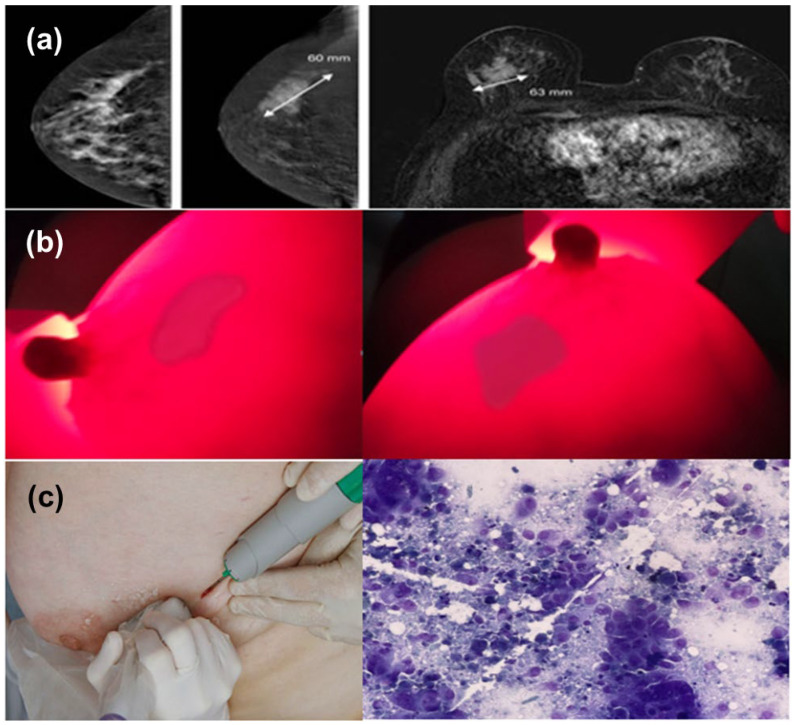
Images of (**a**) MRI scans, (**b**) BKA-06, and (**c**) cell biopsies of breast cancer patient no. 6.

**Figure 10 bioengineering-10-01272-f010:**
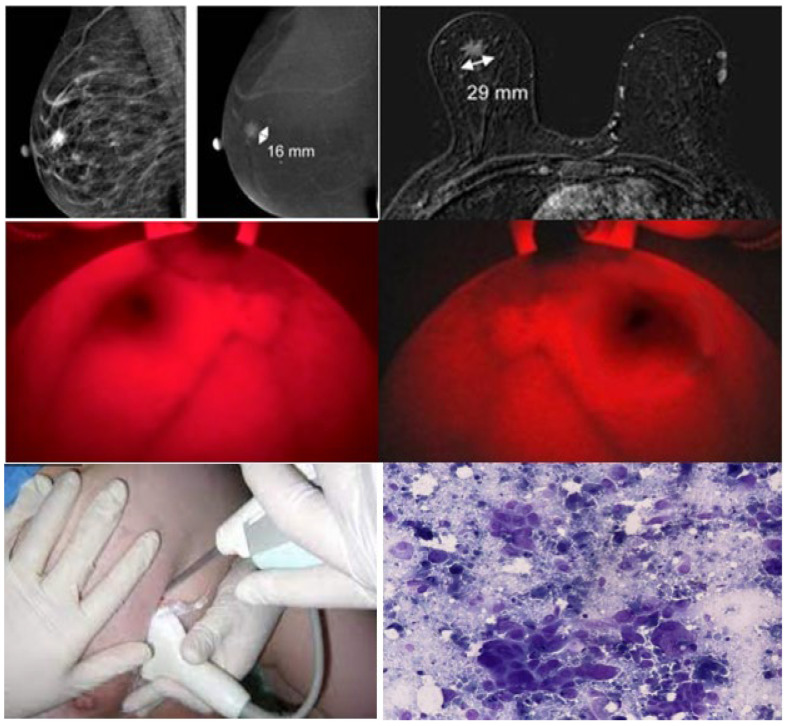
Images of MRI scans, BKA-06, and cell biopsies of breast cancer patient no. 7.

**Figure 11 bioengineering-10-01272-f011:**
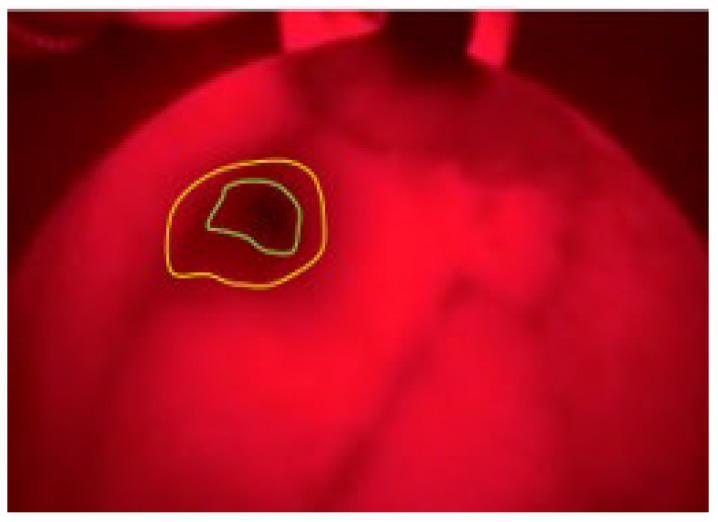
BKA-06 image of patient no. 7.

**Figure 12 bioengineering-10-01272-f012:**
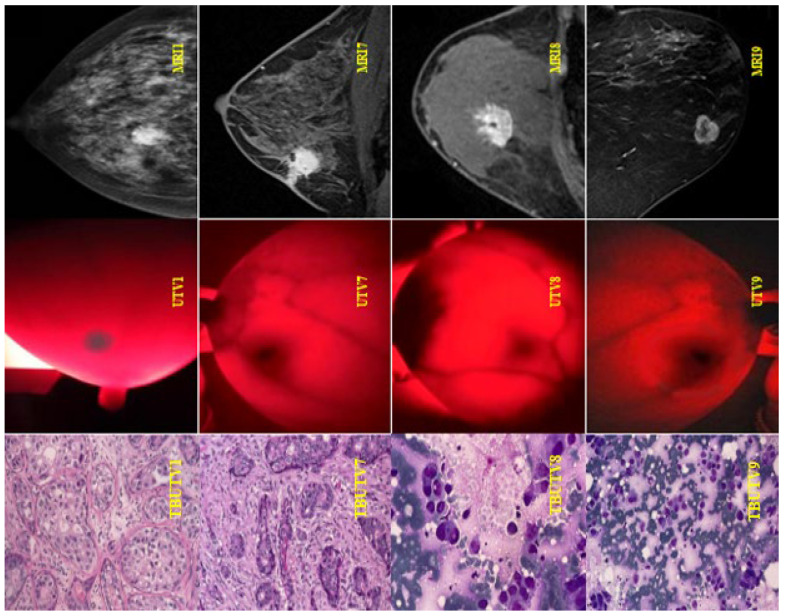
Images of MRI scans, BKA-06, and cell biopsies of breast cancer patients.

**Table 1 bioengineering-10-01272-t001:** Results of measuring the device power and illuminance.

Amperage I (A)	Voltage U (V)	Power P (W)	Light Intensity (W/m^2^)
3.02	0.75	2.27	5.03
3.06	0.82	2.50	6.07
3.14	0.90	2.82	7.70
3.26	1.00	3.25	10.24
3.35	1.10	3.68	13.16
3.45	1.21	4.17	17.07
3.55	1.33	4.73	21.73
3.65	1.46	5.35	28.08
3.75	1.61	6.04	35.59
3.85	1.77	6.81	45.22
3.95	1.94	7.67	57.33
4.05	2.13	8.63	72.49
4.15	2.33	9.69	91.31
4.25	2.56	10.87	114.99
4.35	2.80	12.19	143.93

**Table 2 bioengineering-10-01272-t002:** List of volunteers for breast cancer screening (3 April–28 May 2022).

No.	Full Name	Age	Code
1	Volunteer 1	43	CV1
2	Volunteer 2	31	CV2
3	Volunteer 3	28	CV3
4	Volunteer 4	41	CV4
5	Volunteer 5	55	CV5
6	Volunteer 6	30	CV6
7	Volunteer 7	60	CV7
8	Volunteer 8	40	CV8
9	Volunteer 9	65	CV9
10	Volunteer 10	40	CV10
11	Volunteer 11	18	CV11
12	Volunteer 12	42	CV12
13	Volunteer 13	36	CV13
14	Volunteer 14	25	CV14
15	Volunteer 15	58	CV15

(In this table, some volunteers requested not to publish their phone number, the last two numbers of the symbol XX, and the image of their breasts is committed to each individual to be allowed to publish the images and the information in scientific journals. They volunteered to participate for scientific purposes, not for any other purposes).

**Table 3 bioengineering-10-01272-t003:** List of breast cancer test patients (April–December 2022).

Number	Full Name	Age	Encrypted Image
1	Patient 1	25	UHV1–MRI1
2	Patient 2	67	UHV1–MRI2
3	Patient 3	45	UHV3–MRI3
4	Patient 4	65	UHV4–MRI4
5	Patient 5	45	UHV5–MRI5
6	Patient 6	58	UHV6–MRI6
7	Patient 7	64	UHV7–MRI7
8	Patient 8	52	UHV8–MRI8
9	Patient 9	46	UHV9–MRI9
10	Patient 10	43	UHV10–MRI10
11	Patient 11	59	UHV11–MRI11
12	Patient 12	58	UHV12–MRI12

## Data Availability

Data will be provided upon requested from the authors.

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
