# Peer review of "Fabrication of a Reflective Optical Imaging Device for Early Detection of Breast Cancer"

_bioengineering, 2023, doi:10.3390/bioengineering10111272_

Round 1

Reviewer 1 Report

Comments and Suggestions for Authors

The manuscript presents the fabrication of an optical imaging device based on red light (633 nm) reflection from the tissue. The device was tested on breast imaging of 15 health subjects and 12 patients.  The manuscript lack important components as a scientific paper. The working principle, optical layout of the device is not presented. The title is not accurate. The device is actually a reflectance optical imaging device. It uses only one dominant wavelength (633 nm), no spectroscopy is involved. So the title "optical energy spectroscopy-based device" is very misleading. Major improvement is needed for publication in any journals. Below are some more specific comments:

1) Introduction section.  There are great amount of work on using optical reflectance to image breast and to detect blood vessels in the literature. These work should be reviewed and then outline the differences/improvements of the authors' work. 

2) The illumination is presented using lux. It will be better to be "power density". And the illumination power density has to be compared with safety standards to make sure that the optical power level used is safe for the patient. The authors mentioned that the breast was heated up by the red light illumination and they detected that by thermal imaging. There is really a safety concern. 

3) The study involves human subjects. But there is no IRB review and approval. There is no informed consent.  Subject name and address are disclosed in the paper. These are ethically unacceptable. 

4) page 3, line 107. "G factor" should be "g factor".

5) Page 6, 1st paragraph, it mentioned endoscopy exam of breast. This is a bit confusing. How could endoscopy be used here? More detailed explanation is needed.

6) Fig 7 to fig 15, what camera was used to take the optical images shown? Should label which images are MRI or mammography.

7) Fig 11 is missing.

8) Page 14, line 272.  It mentioned 3D images. How are 3D imaging acquired? what's the working principle? These should be described in section 2.  

Comments on the Quality of English Language

There are numerous grammar errors through out the manuscript.

Author Response

1) Introduction section.  There are great amount of work on using optical reflectance to image breast and to detect blood vessels in the literature. These work should be reviewed and then outline the differences/improvements of the authors' work. 

Response: Thank you very much for the valuable comment. The works on the use of optical reflectance to image breast have been outlined and discussed in the introduction part. The novelty of the work was also clearly stated in the introduction part. Please see the yellow highlight in the manuscript.

2) The illumination is presented using lux. It will be better to be "power density". And the illumination power density has to be compared with safety standards to make sure that the optical power level used is safe for the patient. The authors mentioned that the breast was heated up by the red light illumination and they detected that by thermal imaging. There is really a safety concern. 

Response: Thank you very much for the comment. The power density used for the device was Lithium ion battery of 9V, which is relatively safe for the device application. This was added. Please see the yellow highlight in the manuscript.

3) The study involves human subjects. But there is no IRB review and approval. There is no informed consent.  Subject name and address are disclosed in the paper. These are ethically unacceptable. 

Response: Thank you very much for the valuable comment. Actually, the testing on human had been reviewed and approved by the Medical Ethics Council of hospitals under the Vietnam Ministry of Health. We added this information. The subject name and address were also removed from the paper. Please see the yellow highlight in the manuscript.

4) page 3, line 107. "G factor" should be "g factor".

Response: Thanks. The error was corrected.

5) Page 6, 1st paragraph, it mentioned endoscopy exam of breast. This is a bit confusing. How could endoscopy be used here? More detailed explanation is needed.

Response: Thank you very much for the valuable comment. It was our misleading claim. In reality, it's reflectance imaging rather than endoscopy. (Reflectance imaging with the BKA-06 device uses LED illumination, similar to flashlight illumination, to aid in observing tumors under the skin's surface due to the optical absorption properties of the tumor at specific wavelengths. Photography is used to capture the observed images.). This information was corrected. Please see the yellow highlight in the manuscript.

6) Fig 7 to fig 15, what camera was used to take the optical images shown? Should label which images are MRI or mammography.

Response: Thank you very much for the comment. The camera used was the Sony Alpha ILCE-6400L/A6400 Kit 16-50mm F3.5-5.6 OSS Camera (additional information provided in line 217 on page 6) to capture images in Figures 10, 11, and 13 in order. The first image is a black moon image representing MRI, followed by the image captured using BKA-06 reflectance optical imaging, and finally, the image of cell sampling and biopsy.

7) Fig 11 is missing.

Response: Thank you for the comment. There were misleading in label of the Figures. We corrected the order of the Figures. Please see the yellow highlight in the manuscript.

8) Page 14, line 272.  It mentioned 3D images. How are 3D imaging acquired? what's the working principle? These should be described in section 2.  

Response: Thank you for the comment. This was our misleading claim as the device only obtains the 2D images. We corrected this mistake. Please see the yellow highlight in the manuscript.

Reviewer 2 Report

Comments and Suggestions for Authors

The work is novel and practical, the design and fabrication BKA-06 is interesting and meaningful, but the typesetting of the draft is a bit confusing, and the expression is not standardized, I’d suggest its publication after the minor revisions:

1.     I’d suggest hide all the name of patients or volunteers in the manuscript, I worried it is a violation of publishing ethics, I need to confirm with the editor.

2.     Line 174&176, what’s the meaning of 3/7 and 7/15? (Check other places that has the similar statements) Please draw the scale bar in Figure 7a.

3.     Why not combine Fig. 7a-7e together?

4.     Line 180, “on the hands” should be “on the arm and hands”.

5.     The arrangement of Fig. 7e is wired and what’s the model in B1&B2?

6.     Line 203 should be “Fig. 8, table 3, and Fig. 9”.

Comments on the Quality of English Language

The English need to be revised broadly.

Author Response

  1. I’d suggest hide all the name of patients or volunteers in the manuscript, I worried it is a violation of publishing ethics, I need to confirm with the editor.

Response: Thank you very much for the valuable comment. All the names and address of the patients were removed. Please see the manuscript for the correction.

  1. Line 174&176, what’s the meaning of 3/7 and 7/15? (Check other places that has the similar statements) Please draw the scale bar in Figure 7a.

Response: Thank you for the comment. It was 3 to 7 cm and 7 to 15 cm. It was corrected. Please see the yellow highlight in the manuscript.

  1. Why not combine Fig. 7a-7e together?

Response: Thank you very much indeed for the recommendation. The Figure 7a-7e was combined. Please see the yellow highlight in the manuscript.

  1. Line 180, “on the hands” should be “on the arm and hands”.

Response: Thank you. The grammar errors were corrected. Please see the yellow highlight in the manuscript.

  1. The arrangement of Fig. 7e is wired and what’s the model in B1&B2?

 Response: Thank you for the comment. The B1 and B2 was obtained from the optical microscope. However, B1 and B2 images were removed from the Figure 7e for the scientific logic and sound. Please see the correction in the manuscript.

  1. Line 203 should be “Fig. 8, table 3, and Fig. 9”.

Response: Thank you for the comment. The error was corrected. Please see the yellow highlight in the manuscript.

Reviewer 3 Report

Comments and Suggestions for Authors

Dear respected authors,

I appreciate the valuable insights presented in your manuscript entitled "Development and Testing of the BKA-06 Device for Vein Identification and Breast Cancer Diagnosis". Your research holds significant promise in the field of medical diagnostics, and with some revisions, we believe your manuscript can reach its full potential.

Only a few minor comments:

Introduction:

Acknowledge the concise background provided, but consider further condensing it to highlight the most crucial context.

Explicitly emphasize the novelty and significance of the BKA-06 device in the realm of medical diagnostics (more concise and “to the point” introduction is preferred).

Discussion:

Extensively explore the ramifications of the results, particularly underscoring the unique attributes and contributions of the BKA-06 device in comparison to existing diagnostic methods.

Delve into the practical advantages, potential limitations, and any pertinent safety considerations associated with the device's application.

Future Work:

Consider adding a section on future work, discussing potential areas for further research and development of the device.

Clinical Applicability:

Consider adding a section that discusses the current readiness and potential challenges associated with integrating the BKA-06 device into clinical practice. Highlight any regulatory or practical hurdles that may need to be addressed before widespread adoption.

References:

Ensure that your references are up-to-date and include the most recent studies and developments in the field of medical diagnostics.

Author Response

Introduction: Acknowledge the concise background provided, but consider further condensing it to highlight the most crucial context.Explicitly emphasize the novelty and significance of the BKA-06 device in the realm of medical diagnostics (more concise and “to the point” introduction is preferred).

Response: Thank you very much indeed for the valuable comment. The works on the use of optical reflectance to image breast have been outlined and discussed in the introduction part. The novelty of the work was also clearly stated in the introduction part. Please see the yellow highlight in the manuscript.

Discussion: Extensively explore the ramifications of the results, particularly underscoring the unique attributes and contributions of the BKA-06 device in comparison to existing diagnostic methods. Delve into the practical advantages, potential limitations, and any pertinent safety considerations associated with the device's application.

Response: Thank you very much indeed for the valuable comment. The discussion on the unique attributes and contributions of the BKA-06 device in comparison to existing diagnostic methods and delve into the practical advantages were added into the discussion part. Please see the yellow highlight into the manuscript.

Future Work: Consider adding a section on future work, discussing potential areas for further research and development of the device.

Response: Thank you for the comment. The future research development on the device was added into the conclusion part. Please see the yellow highlight in the manuscript.

Clinical Applicability: Consider adding a section that discusses the current readiness and potential challenges associated with integrating the BKA-06 device into clinical practice. Highlight any regulatory or practical hurdles that may need to be addressed before widespread adoption.

Response: Thank you for the comment. The future research development on the device was added into the conclusion part. Please see the yellow highlight in the manuscript

References: Ensure that your references are up-to-date and include the most recent studies and developments in the field of medical diagnostics.

Response: Thank you for the comment. The references were carefully checked followed the formatted, and the up-to-dated references were also added and discussed.

Round 2

Reviewer 1 Report

Comments and Suggestions for Authors

I have reviewed the authors responses to my reviews. 

First of all, they did not reply to my general comments in the 1st paragraph. There are way more important questions that need to be answered than those 8 specific questions. I am repeating these comments here again:

"The manuscript lack important components as a scientific paper. The working principle, optical layout of the device is not presented. The title is not accurate. The device is actually a reflectance optical imaging device. It uses only one dominant wavelength (633 nm), no spectroscopy is involved. So the title "optical energy spectroscopy-based device" is very misleading." These have to be addressed.

Below are my comments to their response on the 8 specific questions. 

1) The authors did not answer my questions properly. They did not realize that their BK06 device is actually a CW light based reflectance/transmission imaging based device. So they have to compare the BK06 device with existing CW light based reflectance/transmission devices and point out any novelties the BK06 device might have.

2) The authors did not answer my question at all. My question has nothing to do with what batteries they use to power the light source. I was asking them to disclose the light intensity using the unit of optical power density (e.g. in unit of W/cm2, not lux).  They have to compare this power density with the skin safe exposure standard to convince people that their device is safe to the patients. They have indicated that the breast was heated up by the red light illumination and they detected that by thermal imaging. There is really a safety concern that has to be addressed.

3) This question has been addressed. But the ethics certificate number should be disclosed in the manuscript. 

4) This question has been fully addressed.

5) ok.

6) ok, but MRI, BK-6 image etc. should be labelled on the actual images. 

7) ok.

8) ok

Comments on the Quality of English Language

None.

Author Response

  1. "The manuscript lack important components as a scientific paper. The working principle, optical layout of the device is not presented.

Response:  We sincerely thank the reviewers for their valuable comments. We present the operating principle of the device in detail from line 116, page 3 to line 143, page 4. We have added the optical layout of the device in figure 5, page 5.

  1. The title is not accurate. The device is actually a reflectance optical imaging device. It uses only one dominant wavelength (633 nm), no spectroscopy is involved. So the title "optical energy spectroscopy-based device" is very misleading." These have to be addressed.

Response: We sincerely thank the reviewers for their valuable comments. We have revised the name of the article to match the operating principles of the device as suggested by the reviewer. The new name is: “Fabrication of a reflective optical imaging device for early detection of breast cancer”

  1. The authors did not answer my questions properly. They did not realize that their BK06 device is actually a CW light based reflectance/transmission imaging based device. So they have to compare the BK06 device with existing CW light based reflectance/transmission devices and point out any novelties the BK06 device might have.

Response:  We sincerely thank the reviewers for their valuable comments. We have found that our BK06 device belongs to the category of CW light based reflectance/transmission devices. We have compared and drawn out the outstanding advantages of the device compared to existing ones with the same type. The changes have been highlighted from line 75, page 2 to line 107, page 3.

  1. The authors did not answer my question at all. My question has nothing to do with what batteries they use to power the light source. I was asking them to disclose the light intensity using the unit of optical power density (e.g. in unit of W/cm2, not lux).  They have to compare this power density with the skin safe exposure standard to convince people that their device is safe to the patients. They have indicated that the breast was heated up by the red light illumination and they detected that by thermal imaging. There is really a safety concern that has to be addressed.

Response: We sincerely thank the reviewers for their valuable comments. According to reviewers' comments, we used an energy measuring device to investigate the light intensity in units of W/m2. The changes are shown in Table 1, page 5 and lines 200-208, page 6.

Reviewers' concerns about skin safety are valid. The BK06 device emits light at a wavelength of 633nm, near infrared, completely safe for human skin. Survey results showed that the patient's breast heated up when irradiated with red light. However, the temperature change ΔT°C < 1°C is very small to be harmful to the patient's skin (lines from 226-228, page 7).

  1. This question has been addressed. But the ethics certificate number should be disclosed in the manuscript. 

Response: We sincerely thank the reviewers for their valuable comments. The measurement results are legally certified by the Vietnam Metrology Institute, number V11.CN6.3000.23. Information is added in lines 175,176 on page 5.

  1. ok, but MRI, BK-6 image etc. should be labelled on the actual images. 

Response: We sincerely thank the reviewers for their valuable comments. The images have been labeled onto the actual images. These changes are shown in Fig. 10 and caption page 11.